# Personal values in adolescence and psychological distress in adults: A cross-sectional study based on a retrospective recall

**Naonori Yasuma[1,2], Kazuhiro Watanabe[1], Mako Iida[3], Daisuke Nishi[1], Norito Kawakami**[1]*

**1** Department of Mental Health, Graduate School of Medicine, The University of Tokyo, Bunkyo-ku, Tokyo, Japan, **2** Department of Community Mental Health and Law, National Institute of Mental Health, National Center of Neurology and Psychiatry, Kodaira, Tokyo, Japan, **3** Department of Psychiatric Nursing, Graduate School of Medicine, The University of Tokyo, Bunkyo-ku, Tokyo, Japan

* nkawakami@m.u-tokyo.ac.jp

## Abstract

### Background

The aim of this study is to examine the relationship between personal values in adolescence retrospectively assessed and psychological distress in adults in a representative sample of community adults in Japan.

### Methods

We used the J-SHINE data collected in 2010 and 2017. Personal values in adolescence were retrospectively measured in the 2017 survey in two ways: (1) value priorities developed from Schwartz's theory of basic values; and (2) commitment to values measured by Personal Values Questionnaire II (PVQ-II). Psychological distress was measured by using K6 in 2010. Multiple regression analysis was conducted to clarify the association between personal values in adolescence and psychological distress in adults, adjusting for sociodemographic variables, smoking, alcohol drinking, and economic status at age 15.

### Results

Enduring active challenging, cherishing family and friends, and the commitment to values were significantly and negatively associated with psychological distress in adults. Pursuing one's interest was significantly and positively associated with psychological distress.

### Conclusions

Having a value priority of enduring active challenging, cherishing family and friends, and the commitment to values in adolescence may reduce psychological distress in adults.

**Data Availability Statement:** Data are available from the Data Committee of the Japanese Study on Stratification, Health, Income, and Neighborhood (J-SHINE) for researchers who meet the criteria for

access to data. The ethical approval of the J-SHINE study was obtained with a clear statement that the data would be used with the permission of the Data Management Committee. The Data Management Committee is concerned that sharing data, even when these are anonymous, might happen to result in identifying some of the respondents, if it is not well controlled. We cannot share the data without the permission of the Data Management Committee, according to our research plan submitted to the ethics committee at the University of Tokyo. Data access requests may be directed to Prof Hideki Hasimoto, the Chair of the Data Management Committee of the JSHINE study, at hidehasimoto-circ@umin.ac.jp.

**Funding:** This study was partly supported by MEXT KAKENHI Grant Number JP21119003, JSPS KAKENHI Grant Number JP16H06395, 16H06398, and 16K21720. The Japanese Study on Stratification, Health, Income, and Neighborhood (J-SHINE) was supported by a Grant-in-Aid for Scientific Research on Innovative Areas (No. 1119002) from the Ministry of Education, Culture, Sports, Science and Technology, Japan. The funders had no role in the study design, data collection and analysis, the writing of the report, or the decision to submit the paper for publication.

**Competing interests:** The authors have declared that no competing interests exist.

## Introduction

Personal value is a broad goal, varying in importance and underlying and guiding attitudes and behavior [1, 2]. Personal value is divided into two major components: the content of values and the commitment to those values [3, 4]. The content of values are the things people value, such as power, achievement, hedonism, stimulation, self-direction, universalism, benevolence, tradition, conformity, and security [3]. The commitment to values represents how much a person emphasizes and acts according to those values [5].

Personal values are formed in adolescence, which may affect long-term cognitions, behaviors, and finally health and well-being [1, 2]. Previous studies found that one's personal values in adolescence, both the content of and commitment to those values, were associated with positive mental well-being [6, 7]. However, no study has clarified the associations between personal values in adolescence and psychiatric symptoms, such as depression and anxiety. Investigating this association is important because it could lead researchers to know if personal values could improve mental health and prevent mental disorders in adulthood.

The purpose of this study is to examine the association between personal values in adolescence retrospectively assessed and psychological distress (depression and anxiety) in a large representative sample of community-residing adults in Japan.

## Materials and methods

### Participants

The study protocol was approved by the Research Ethics Committee of the Graduate School of Medicine and the Faculty of Medicine, The University of Tokyo, Japan [No.630-7,3361]. This study was a cross-sectional study based on a retrospective recall by using wave 1 and wave 3 data of the Japanese Study on Stratification, Health, Income, and Neighborhood (J-SHINE) survey [8]. The sample was randomly selected from adult residents aged 20 to 50 years from four municipalities (two in Tokyo; two in neighboring prefectures). Systematic sampling methods were used from a residents' register. There were no inclusion and exclusion criteria except for age. The participants were received invitation letters, and trained surveyors visited their houses. The participants were also asked to provide written informed consent and answered the self-administered questionnaire with a computer-aided personal instrument (CAPI). Three investigations have been conducted to date (Fig 1). In the wave 1 survey in 2010, 4,357 people completed the questionnaire (response rate: 31.3%). In the wave 2 survey in 2012, 2,961 people who had responded to the survey in wave 1 were recruited (response rate: 69.0%). In the wave 3 survey in 2017, those who responded to both the wave 1 and wave 2 surveys were selected; 2,787 people answered the questionnaire (response rate: 64.9%). Psychological distress and socio-demographics were measured in wave 1 and personal values in adolescence were measured in wave 3.

### Personal values in adolescence

We measured personal values in adolescence by using the value priorities and the commitment to values. For the value priorities, we developed 11 items based on the 57- item Portrait Values Questionnaire (PVQ-57; [9]) to be suitable to assess value priorities in adolescence. First, based on Schwartz's 10 motivationally distinct priorities, we selected and revised eight items from the PVQ-57: avoiding causing trouble, positive evaluation, belief, improving society, social influence, enduring active challenging, cherishing family and friends, and stable lifestyle. Then, we developed three additional items that could motivate behavior, especially in adolescence: financial success, pursuing one's interest, and graduating from school. These items are

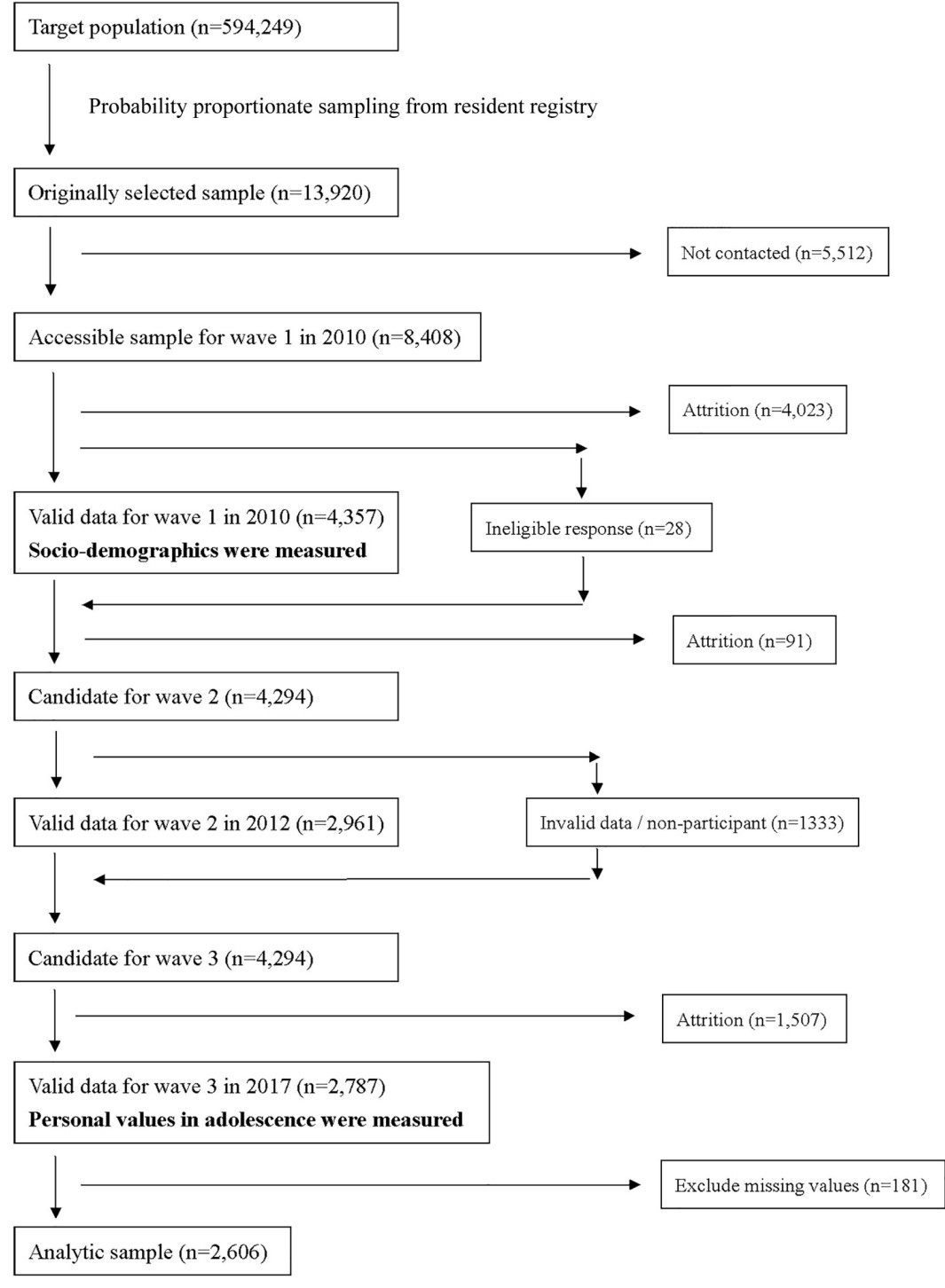

**Fig 1. Flowchart of participant recruitment in J-SHINE.**

rated on a seven-point Likert scale (1 = Not at all, 7 = Very important) following the question, "When you were 15–16 years old, how important did you think the following values were in your life?" The commitment to values was measured by the Japanese version of the Personal Values Questionnaire II (PVQ-II) [5]. The reliability and validity of Japanese version of

PVQ-II has already been confirmed. In this study, we revised the items to the past tense and instructed the participants to answer the items they considered the most important when they were 15–16 years old.

### Psychological distress

We used Japanese version of K6 for measurement of psychological distress (depression and anxiety) [10]. It consists of six items answered on a 5-point Likert scale. Higher scores represent higher degrees of psychological distress. The reliability and validity of Japanese version of K6 has already been confirmed.

### Data analysis

Multiple regression analysis was used to estimate the association between personal values in adolescence and psychological distress in adults. Model 1 was adjusted for socio-demographics variables; model 2 was adjusted for smoking and drinking alcohol; model 3 was adjusted for economic status at age 15. A $p$-value less than 0.05 was statistically significant. We used SPSS (Windows version 25) for statistical analysis. Imputation of missing responses on the variables was not conducted.

## Results

Among 2,787 total survey respondents, 2,723 completed the value priorities and PVQ-II. Some of the respondents had missing values on K6, demographic variables, smoking, drinking alcohol and economic status in 15 years old (n = 117) and were excluded from the study. The final sample that had no missing values and was used for analysis comprised 2,606 respondents (Fig 1). S1 Table showed demographic and psychosocial characteristics of the study. According to S2 Table, pursuing one's interest was significantly and positively associated with psychological distress after adjusting for socio-demographics variables (model 1), smoking and drinking alcohol (model 2) and economic status in 15 years old (model 3). Enduring active challenging was significantly and negatively associated with psychological distress after adjusting for model 2 and model 3, however, there were no significant associations after adjusting for model 1. Cherishing family and friends and the commitment to values were significantly and negatively associated with psychological distress after adjusting for model 1, model 2 and model 3.

## Discussion

The results revealed that enduring active challenging, cherishing family and friends and the commitment to values were significantly and negatively associated with psychological distress in adults; pursuing one's interest was significantly and positively associated with psychological distress.

Considering Schwartz's 10 basic values theory, previous studies revealed that values related to openness to change (self-direction, stimulation and hedonism), achievement, and benevolence were positively associated with mental health and well-being [6][11][12]. In this study, enduring active challenging and cherishing family and friends were almost the same concepts as stimulation, self-direction and benevolence, respectively. Enduring active challenging and cherishing family and friends were consistent with the previous studies, however, pursuing one's interest was opposite to previous findings. Considering possible reasons for this finding, pursuing one's interest might be considered less socially acceptable in Japan, because the Japanese believe that harmony is the greatest of virtues [13].

Previous studies have indicated that the commitment to values was associated with well-being [7], which showed that this study was consistent with the literature. Those who committed themselves to their own values may have created high self-efficacy and/or self-esteem, which could decrease psychological distress in adults.

This study has two strengths. One is that the topic of personal values in adolescence had not yet been fully researched before this study. The other is that, as a practical implication, moral education to enduring active challenging and cherishing family and friends in adolescence could improve mental health in adulthood.

However, there were five limitations. First, recall bias may have occurred because the participants had to remember the values that were important to them when they were 15 years old. Second, childhood adversity should be taken into account as a confounding factor because it affected psychological distress and was presumably associated with personal values in adolescence. Third, test-retest reproducibility, internal consistency and construct validity of the measurement of value priorities in adolescence used in this study has not been fully evaluated, so random error is a possibility. Fourth, the low response rate of this study could cause the possibility of selection bias, which could also lead to uncertainty in the representativeness and the generalization of the results. Fifth, as this study was cross-sectional, so a cohort study should be conducted to clarify the causality of this association.

## Conclusions

Despite these limitations, the present study showed that enduring active challenging, cherishing family and friends and the commitment to values in adolescence could help to reduce psychological distress in adults.

## Supporting information

**S1 Table. Demographics and psychosocial characteristics of the participants.**
(DOCX)

**S2 Table. Personal values in adolescence and psychological distress in adulthood: Multiple Linear regression analysis of data from community residents in Japan.**
(DOCX)

## Acknowledgments

The authors would like to thank the data control committee of the Japanese Study of Stratification, Health, Income, and Neighborhood (J-SHINE) research group for providing us the data. We also extend our great appreciation to Professor Hideki Hashimoto and Lecturer Daisuke Takagi for their help in data handling.

## Author Contributions

**Conceptualization:** Naonori Yasuma.

**Data curation:** Naonori Yasuma.

**Formal analysis:** Naonori Yasuma.

**Funding acquisition:** Norito Kawakami.

**Investigation:** Naonori Yasuma.

**Methodology:** Naonori Yasuma.

**Project administration:** Naonori Yasuma, Norito Kawakami.

**Supervision:** Kazuhiro Watanabe, Daisuke Nishi, Norito Kawakami.

**Writing – original draft:** Naonori Yasuma.

**Writing – review & editing:** Kazuhiro Watanabe, Mako Iida, Daisuke Nishi.

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
