## [Editor Report · Decision Letter 0]

22 Oct 2019

PONE-D-19-19474

Brief report: Personal values in adolescence and psychological distress in adults: A cross-sectional study based on a retrospective recall

PLOS ONE

Dear Dr. Kawakami,

Thank you for submitting your manuscript to PLOS ONE. After careful consideration, we feel that it has merit but does not fully meet PLOS ONE’s publication criteria as it currently stands. Therefore, we invite you to submit a revised version of the manuscript that addresses the points raised during the review process.

We would appreciate receiving your revised manuscript by Dec 06 2019 11:59PM. To enhance the reproducibility of your results, we recommend that if applicable you deposit your laboratory protocols in protocols.io, where a protocol can be assigned its own identifier (DOI) such that it can be cited independently in the future. For instructions see: http://journals.plos.org/plosone/s/submission-guidelines#loc-laboratory-protocols

We look forward to receiving your revised manuscript.

Kind regards,

Geilson Lima Santana, M.D., Ph.D.

Academic Editor

PLOS ONE

**Journal Requirements:**

2. Please provide additional information about the participant recruitment method and the demographic details of your participants.

3. We note that the manuscript title begins with "Brief report". Please note that PLOS ONE does not have an article type of this name. Please consider removing this from your article title."

**Additional Editor Comments (if provided):**

Dear authors,

I believe this is an interesting and relevant article, and may contribute to the understanding of the influence of personal values in adolescence on adulthood mental health. However, before publication, some minor revision is needed:

Regarding data availability, it is not possible to find them in the article, which only contains a regression table. Plos One requires that the data are made available as an attached file or at a public repository. Eventually, data may be offered upon request, due to ethical or privacy issues.

Another important issue if the low response rate in wave 1 (31.3%) – and, therefore, for the whole study, since wave 1 was the basis for the recruitment of all participants in the study. This may have led to a lack of representativeness and limits to the generalization of the results. It is important to make a reflection on this on the discussion. Moreover, since there was such a low response rate in the study, I suggest avoiding the following expression in the discussion: “It was able to avoid selection bias by using systematic sampling”.

Regarding the “value priorities”, you have developed 11 items based on the 57- item Portrait Values Questionnaire (PVQ-57). How was this done? Which procedures were adopted? Which were the criteria for the selection of the items included in the study? Was any psychometric analysis performed?

I would suggest placing the first paragraph of the results on the methods section. It is more related to “participants” than to “results”.
---

## [Author Response · Author response to Decision Letter 0]

1 Nov 2019

Dear Editor in chief

Thank you for inviting us to submit a revised manuscript entitled, “Personal values in adolescence and psychological distress in adults: A cross-sectional study based on a retrospective recall” to PLOS ONE. We also appreciate the time and effort you and each of the reviewers have dedicated to providing insightful feedback on ways to strengthen our paper. Thus, it is with great pleasure that we resubmit our article for further consideration. 

We have incorporated changes that reflect the detailed suggestions you have graciously provided. We prepared point-by-point responses to the questions and comments on separate sheets. We hope that our edits and the responses we provide below satisfactorily address all the issues and concerns you and the reviewers have noted.

Again, thank you for giving us the opportunity to revise and strengthen our manuscript with your valuable comments and queries. We look forward to your positive decision on our submission.

Sincerely,

Norito Kawakami

Reply to Journal Requirements

Thank you very much for your valuable and helpful suggestions. We have revised the manuscript in accordance with your suggestions. Revisions are shown in highlighted text in the revised manuscript. We would be pleased if you could have a look at the revised manuscript and check if we responded to your comments appropriately.

Comment 1

When submitting your revision, we need you to address these additional requirements. Please ensure that your manuscript meets PLOS ONE's style requirements, including those for file naming. The PLOS ONE style templates can be found at 

http://www.journals.plos.org/plosone/s/file?id=wjVg/PLOSOne_formatting_sample_main_body.pdf

and

http://www.journals.plos.org/plosone/s/file?id=ba62/PLOSOne_formatting_sample_title_authors_affiliations.pdf

Response: Thank you very much for your comment. We revised our manuscript in accordance with PLOS ONE’s style.

Comment 2

Please provide additional information about the participant recruitment method and the demographic details of your participants.

Response: Thank you very much for your comment. We added the participant recruitment method in the manuscript and also explained with a newly added figure (Fig 1). We also created a new table concerning about demographic details of the participants (Table 1).

The sample was randomly selected from adult residents aged 20 to 50 years from four municipalities (two in Tokyo; two in neighboring prefectures). Systematic sampling methods were used from a residents’ register. There were no inclusion and exclusion criteria except for age. The participants were received invitation letters, and trained surveyors visited their houses. The participants were also asked to provide written informed consent and answered the self-administered questionnaire with a computer-aided personal instrument (CAPI). Three investigations have been conducted to date (Fig 1). (p3, line25-32) 

Comment 3

We note that the manuscript title begins with "Brief report". Please note that PLOS ONE does not have an article type of this name. Please consider removing this from your article title."

Response: Thank you very much for your comment. We removed "Brief report" from the title.

Personal values in adolescence and psychological distress in adults: A cross-sectional study based on a retrospective recall (p1, line1-2)

Reply to Additional Editor Comments 

Thank you very much for your valuable and helpful suggestions. We have revised the manuscript in accordance with your suggestions. Revisions are shown in highlighted text in the revised manuscript. We would be pleased if you could have a look at the revised manuscript and check if we responded to your comments appropriately.

Comment 1

Regarding data availability, it is not possible to find them in the article, which only contains a regression table. Plos One requires that the data are made available as an attached file or at a public repository. Eventually, data may be offered upon request, due to ethical or privacy issues.

Response: Thank you very much for your comment. We added the following sentences in the manuscript.

Data Availability Statement 

Data are available from the Data Committee of the Japanese Study on Stratification, Health, Income, and Neighborhood (J-SHINE) for researchers who meet the criteria for access to data. The ethical approval of the J-SHINE study was obtained with a clear statement that the data would be used with the permission of the Data Management Committee. The Data Management Committee is concerned that sharing data, even when these are anonymous, might happen to result in identifying some of the respondents, if it is not well controlled. We cannot share the data without the permission of the Data Management Committee, according to our research plan submitted to the ethics committee at the University of Tokyo. Data access requests may be directed to Prof Hideki Hasimoto, the Chair of the Data Management Committee of the JSHINE study, at hidehasimoto-circ@umin.ac.jp. (p7, line1-12)

Comment 2

Another important issue if the low response rate in wave 1 (31.3%) – and, therefore, for the whole study, since wave 1 was the basis for the recruitment of all participants in the study. This may have led to a lack of representativeness and limits to the generalization of the results. It is important to make a reflection on this on the discussion. Moreover, since there was such a low response rate in the study, I suggest avoiding the following expression in the discussion: “It was able to avoid selection bias by using systematic sampling”.

Response: Thank you very much for your comment. We added the possibility of causing selection bias, the limitation of the representativeness and the generalization of this study. We also removed the following sentences from the manuscript. “It was able to avoid selection bias by using systematic sampling.”

Fourth, the low response rate of this study could cause the possibility of selection bias, which could also lead to uncertainty in the representativeness and the generalization of the results. (p6, line18-20) 

This study has two strengths. One is that the topic of personal values in adolescence had not yet been fully researched before this study. The other is that, as a practical implication, moral education to enduring active challenging and cherishing family and friends in adolescence could improve mental health in adulthood. (p6, line7-10)

Comment 3

Regarding the “value priorities”, you have developed 11 items based on the 57- item Portrait Values Questionnaire (PVQ-57). How was this done? Which procedures were adopted? Which were the criteria for the selection of the items included in the study? Was any psychometric analysis performed?

Response: Thank you very much for your comment. We added the following explanation and limitation of the measurement tool to the manuscript.

For the value priorities, we developed 11 items based on the 57- item Portrait Values Questionnaire (PVQ-57; [9]) to be suitable to assess value priorities in adolescence. First, based on Schwartz’s 10 motivationally distinct priorities, we selected and revised eight items from the PVQ-57: avoiding causing trouble, positive evaluation, belief, improving society, social influence, enduring active challenging, cherishing family and friends, and stable lifestyle. Then, we developed three additional items that could motivate behavior, especially in adolescence: financial success, pursuing one’s interest, and graduating from school. These items are rated on a seven-point Likert scale (1=Not at all, 7=Very important) following the question, “When you were 15-16 years old, how important did you think the following values were in your life?” (p4, line9-19) 

Third, test-retest reproducibility, internal consistency and construct validity of the measurement of value priorities in adolescence used in this study has not been fully evaluated, so random error is a possibility. (p6, line15-18)

Comment 4

I would suggest placing the first paragraph of the results on the methods section. It is more related to “participants” than to “results”.

Response: Thank you very much for your comment. According to the STROBE statement <https://strobe-statement.org/index.php?id=strobe-home>, this information is recommended to listed at the beginning of the results. Therefore, we would like not to change the position of the sentences. We hope that this is acceptable.

---

## [Editor Report · Decision Letter 1]

6 Nov 2019

Personal values in adolescence and psychological distress in adults: A cross-sectional study based on a retrospective recall

PONE-D-19-19474R1

Dear Dr. Kawakami,

We are pleased to inform you that your manuscript has been judged scientifically suitable for publication and will be formally accepted for publication once it complies with all outstanding technical requirements.

With kind regards,

Geilson Lima Santana, M.D., Ph.D.

Academic Editor

PLOS ONE

Additional Editor Comments (optional):

Thank you for considering the indications and suggestions.

Congratulations for your paper.

Good luck!
---

## [Editor Report · Acceptance letter]

12 Nov 2019

PONE-D-19-19474R1 

Personal values in adolescence and psychological distress in adults: A cross-sectional study based on a retrospective recall 

Dear Dr. Kawakami:

I am pleased to inform you that your manuscript has been deemed suitable for publication in PLOS ONE. Congratulations! Your manuscript is now with our production department. 

With kind regards,

on behalf of

Dr. Geilson Lima Santana 

Academic Editor

PLOS ONE